# Population-Specific *ACE2* Single-Nucleotide Polymorphisms Have Limited Impact on SARS-CoV-2 Infectivity In Vitro

**DOI:** 10.3390/v13010067

**Published:** 2021-01-06

**Authors:** Mei Hashizume, Gabriel Gonzalez, Chikako Ono, Ayako Takashima, Masaharu Iwasaki

**Affiliations:** 1Laboratory of Emerging Viral Diseases, International Research Center for Infectious Diseases, Research Institute for Microbial Diseases, Osaka University, 3-1 Yamadaoka, Osaka 565-0871, Japan; hashizumei@biken.osaka-u.ac.jp (M.H.); atakashima@biken.osaka-u.ac.jp (A.T.); 2UCD National Virus Reference Laboratory, Belfield Campus, Dublin 4, Ireland; gabo.gonzalez@ucd.ie; 3Department of Molecular Virology, Research Institute for Microbial Diseases, Osaka University, 3-1 Yamadaoka, Osaka 565-0871, Japan; chikaono@biken.osaka-u.ac.jp

**Keywords:** SARS-CoV-2, COVID-19, ACE2, SNPs, viral cell entry

## Abstract

Severe acute respiratory syndrome coronavirus 2 (SARS-CoV-2), the causative agent of coronavirus disease 2019 (COVID-19), employs host-cell angiotensin-converting enzyme 2 (ACE2) for cell entry. Genetic analyses of *ACE2* have identified several single-nucleotide polymorphisms (SNPs) specific to different human populations. Molecular dynamics simulations have indicated that several of these SNPs could affect interactions between SARS-CoV-2 and ACE2, thereby providing a partial explanation for the regional differences observed in SARS-CoV-2 infectivity and severity. However, the significance of population-specific *ACE2* SNPs in SARS-CoV-2 infectivity is unknown, as no in vitro validation studies have been performed. Here, we analyzed the impact of eight SNPs found in specific populations on receptor binding and cell entry in vitro. Except for a SNP causing a nonsense mutation that reduced ACE2 expression, none of the selected SNPs markedly altered the interaction between ACE2 and the SARS-CoV-2 spike protein (SARS-2-S), which is responsible for receptor recognition and cell entry, or the efficiency of viral cell entry mediated by SARS-2-S. Our findings indicate that *ACE2* polymorphisms have limited impact on the ACE2-dependent cell entry of SARS-CoV-2 and underscore the importance of future studies on the involvement of population-specific SNPs of other host genes in susceptibility toward SARS-CoV-2 infection.

## 1. Introduction

A novel coronavirus, severe acute respiratory syndrome coronavirus 2 (SARS-CoV-2), and the associated respiratory disease, coronavirus disease 2019 (COVID-19), emerged in December 2019. COVID-19 has spread rapidly around the world, causing an unprecedented pandemic. As of 18 December 2020, over 72 million cases and 1.6 million deaths have been reported in 222 countries and territories (WHO, https://www.who.int/emergencies/diseases/novel-coronavirus-2019). As with the other closely related coronaviruses, SARS-CoV-2 uses the host-cell angiotensin-converting enzyme 2 (ACE2) as a primary receptor for entry into target cells [1]. ACE2, encoded on the X chromosome, is a type I cell-surface glycoprotein with peptidase activity that contributes toward the control of blood pressure by regulating the renin–angiotensin–aldosterone system [2]. The S1 subunit of the SARS-CoV-2 spike (S) protein (SARS-2-S) recognizes the ACE2 receptor and this facilitates viral attachment to the cell. Viral cell entry also requires S priming by cellular proteases such as transmembrane protease, serine 2 (TMPRSS2) [1], which is achieved by cleavage of the S protein at the S1/S2 and S2′ sites. This process allows the S2 subunit of the S protein to mediate a fusion event between the viral and cellular membranes. This releases the viral ribonucleoprotein complex, which is responsible for viral gene expression and genome replication, into the cell cytoplasm.

Mounting evidence indicates that there are large regional differences in the proportions of COVID-19 cases and deaths among countries [3,4,5,6,7]. In particular, remarkably low numbers of COVID-19 cases and deaths per population have been reported in East Asian countries, including Vietnam, Thailand, China, and Japan, compared with those in the USA and European countries such as Italy, Spain, the UK, and France (WHO, https://covid19.who.int). One possible explanation for this is that host genetic differences affect SARS-CoV-2 infectivity, thereby contributing toward differences in the morbidity and mortality of COVID-19 among different populations. Knowledge of host genetic variances causing differences in COVID-19 severity, if present, should be beneficial in clinical settings, because this genetic information would help to determine the therapeutic strategies for COVID-19 patients. To address this issue, several groups have investigated single nucleotide polymorphisms (SNPs) in the coding region of *ACE2*, which encodes the primary receptor for SARS-CoV-2, to identify SNPs prevalent in specific populations and to determine their potential impact on ACE2 binding of SARS-2-S. For example, using the FireDock web server-based protein–protein interaction simulator [8], Calcagnile et al. reported that S19P (a SNP common in African populations) and K26R (a SNP common in European populations) may decrease and increase the binding affinity of ACE2 toward SARS-2-S, respectively [9]. However, these findings are controversial, as another group has reported that K26R and I468V (a SNP common in East Asian populations) slightly decrease the binding affinity based on a molecular dynamics simulation using GROMACS 2018.4 [10]. This emphasizes the need for in vitro validation studies to determine the significance of the contribution of population-specific SNPs of *ACE2* toward SARS-CoV-2 infectivity. In this study, we examined the impact of eight *ACE2* SNPs found in specific populations on the receptor binding and cell entry of SARS-CoV-2. Except for a SNP causing a nonsense mutation that diminished ACE2 protein expression, our findings indicate that the selected *ACE2* SNPs have limited impact on the ACE2-mediated cell entry of SARS-CoV-2 and are, therefore, not associated with the regional variations in the severity of COVID-19. This underscores the importance of future studies to assess the involvement of population-specific SNPs of other host factors, such as TMPRSS2, in susceptibility toward SARS-CoV-2 infection.

## 2. Materials and Methods

### 2.1. SNP Analysis

The Genome Aggregation Database (gnomAD), the Japanese Multi Omics Reference Panel (jMorp) [11], and the 1000 Genomes Project [12] databases were used to identify population-specific ACE2 SNPs. We selected East Asian and Japanese population-specific SNPs in the coding region that change amino acid residues with an allele frequency higher than 0.001 (0.1%). For comparison, three other SNPs were also incorporated in our analysis. These comprised S19P and K26R, reported to potentially affect ACE2 affinity toward SARS-2-S [9,10], as well as N720D, which is located far from the SARS-2-S binding domain and is, therefore, expected to have little impact on SARS-2-S binding [10]; these three SNPs are common in American and Non-Finnish European populations.

### 2.2. Plasmids

To generate the plasmid (pCMV6-hACE2-FLAG) expressing C-terminally FLAG-tagged wild-type human ACE2 (WT ACE2), a DNA fragment encoding the full-length human ACE2 open reading frame (ORF) was amplified by PCR using a human ACE2-expressing plasmid (HG10108-UT; Sino Biological, Beijing, China) as a template and then inserted in-frame into the pCMV6-Entry vector (OriGene, Rockville, MD, USA) between the SgfI and EcoRV sites connecting the ACE2 ORF and FLAG-tag cDNA sequences. Plasmids expressing human ACE2 variants with a C-terminus FLAG-tag were generated by PCR-based mutagenesis introducing corresponding mutations into pCMV6-hACE2-Flag.

### 2.3. Cells and Viruses

HEK293T cells (CRL-3216; American Type Culture Collection, VA, USA) were cultured at 37 °C and 5% CO_2_ in Dulbecco’s modified Eagle medium (DMEM; Nacalai Tesque, Kyoto, Japan) supplemented with 10% heat-inactivated fetal bovine serum (FBS), 100 U/mL penicillin, and 100 µg/mL streptomycin. HEK293T cells constitutively expressing human ACE2 (SL221; GeneCopoeia, Rockville, MD, USA) were grown at 37 °C and 5% CO_2_ in DMEM containing 10% FBS, 100 U/mL penicillin, 100 µg/mL streptomycin, and 100 μg/mL hygromycin B.

A replication-deficient vesicular stomatitis virus (VSV)-based pseudovirus bearing SARS-2-S with the D614G mutation was generated by following a procedure similar to that previously used to generate a pseudotype recombinant VSV complemented with hepatitis C virus glycoproteins E1 and E2 [13]. HEK293T cells transfected for 24 h with a plasmid expressing the C-terminal 19 amino acid-deleted SARS-2-S (codon-optimized for human cells) containing the D614G substitution (pCAGG-SARS2-S_Hu_∆19-D614G) were inoculated with pseudotyped VSV complemented with the VSV glycoprotein (G), where the G gene was replaced with the green fluorescent protein (GFP) gene [14], at a multiplicity of infection (MOI) of 5. After 2 h adsorption of the virus inoculum, the cells were washed four times with DMEM to remove unbound virions, and fresh medium was added to the cells. After 24 h incubation at 37 °C and 5% CO_2_, the cell culture medium (tissue culture supernatant, TCS) was collected, clarified by centrifugation at 400× *g* at 4 °C for 5 min to remove the cell debris, and the cleared supernatant was stored at −80 °C until use.

### 2.4. Western Blotting

Clarified cell lysates were mixed at a 1:1 ratio with loading buffer (62.5 mM Tris-HCl, pH 6.8, 5% 2-mercaptoethanol, 2% sodium dodecyl sulfate (SDS), 0.01% bromophenol blue, 25% glycerol) and boiled for 5 min. Protein samples were fractionated by SDS-polyacrylamide gel electrophoresis (SDS-PAGE) using 5–20% (Extra PAGE One Precast Gel 5–20%; Nacalai Tesque) or 5–15% [CosmoPAGE TG (Tris-Glycine) Precast Gel 4–15%; Nacalai USA, San Diego, CA, USA] gradient polyacrylamide gels, and the resolved proteins were transferred by electroblotting onto polyvinylidene difluoride membranes (Immobilon-P PVDF Transfer Membranes; Millipore, Burlington, MA, USA). To detect specific proteins, the membranes were incubated with a mouse monoclonal antibody toward ACE2 (AC18F; Cayman Chemical, Ann Arbor, MI, USA), a rabbit polyclonal antibody toward GAPDH (ABS16; Millipore), and rabbit monoclonal antibodies toward the FLAG-tag (PM020; MBL, Aichi, Japan) and the transferrin receptor (H68.4; Invitrogen, Carlsbad, CA, USA), followed by incubation with horseradish peroxidase-conjugated anti-rabbit or anti-mouse immunoglobulin G (IgG) antibodies as appropriate (Jackson ImmunoResearch Laboratories, West Grove, PA, USA). The Chemi-Lumi One L or Chemi-Lumi One Ultra chemiluminescent substrate (Nacalai Tesque) was used to generate chemiluminescent signals that were visualized with the ImageQuant LAS 4000 mini biomolecular imager (GE Healthcare Bio-Sciences, MA, USA). The signal intensities of the visualized protein bands were determined with ImageJ software [15].

### 2.5. Assessment of Cell Surface Expression of ACE2 Variants

HEK293T cells seeded in a six-well plate at 9 × 10^5^ cells per well and cultured overnight were transfected with 1 μg of plasmids expressing WT ACE2 or the ACE2 variants using 2.5 µL of Lipofectamine 2000 (Thermo Fisher Scientific, Waltham, MA, USA). At 24 h posttransfection, the cell-surface proteins on the transfected cells were biotin-labeled with 1 mg/mL Biotin-SS-Sulfo-Osu (Dojindo, Kumamoto, Japan) for 30 min at 4 °C. The biotin-labeled cells were washed twice with ice-cold quenching buffer (50 mM Tris-HCl, pH 7.6 0.1 mM EDTA, 150 mM NaCl) and once with ice-cold phosphate-buffered saline (PBS). The cells were then lysed with 300 µL of 0.3% n-dodecyl-b-D-maltopyranoside (DDM; Dojindo) in PBS supplemented with the Halt Protease and Phosphatase Inhibitor Cocktail (Thermo Fisher Scientific) and incubated for 20 min at 4 °C. The lysate was clarified by centrifugation at 15,000 rpm and 4 °C for 5 min to remove the cell debris (Input samples). The cleared cell lysate was mixed with 20 μL of streptavidin-conjugated magnetic beads (Dynabeads M-280 Streptavidin; Thermo Fisher Scientific) and incubated overnight at 4°C. The beads were washed four times with 0.3% DDM in PBS. Next, 30 μL of loading buffer was added to the beads and the beads were incubated at 37 °C for 30 min, followed by boiling for 5 min. Biotinylated proteins eluted into the loading buffer (PD samples) were separated using a magnetic stand. FLAG-tagged WT ACE2 and ACE2 variant levels in the Input and PD samples were analyzed by western blotting. The ratios of the protein levels on the cell surface to those in the total cell lysates were calculated by dividing the signal intensity of the PD sample by that of the corresponding Input sample.

### 2.6. Assessment of Binding of SARS-CoV-2 S1 Subunit to Cells Expressing ACE2 Variants

HEK293T cells seeded in a 12-well plate at 4 × 10^5^ cells per well and cultured overnight were transfected with 0.2 μg of plasmids expressing WT ACE2 or the ACE2 variants using 0.5 uL of Lipofectamine 2000. At 24 h posttransfection, the transfected cells were collected, mixed with 1.5 µg of the recombinant SARS-CoV-2 S1 subunit containing the D614G mutation fused with the His-tag (Sino Biological), and incubated on ice for 1 h. S1-bound cells were washed with 0.5% bovine serum albumin (BSA) in PBS and incubated for another 30 min on ice with anti-His-tag mouse monoclonal antibody conjugated with Alexa Fluor 488 (D291-A48; MBL) and 7-aminoactinomycin D (7-AAD; Invitrogen) for the detection of dead cells. Cells were then washed with 0.5% BSA in PBS and subjected to flow cytometry analysis with the SH800S Cell Sorter (Sony, Tokyo, Japan) to determine the proportions of live and S1-binding cells.

### 2.7. Virus Entry Assay

HEK293T cells seeded in a 96-well plate at 2 × 10^4^ cells per well and cultured overnight were transfected with 10 ng of plasmids expressing WT ACE2 or the ACE2 variants. At 24 h posttransfection, cells were inoculated with rVSVΔG-GFP/SARS-2-S(D614G) [3 × 10^4^ focus forming units (FFU) per well]. After 16 h, the cells were fixed with 4% paraformaldehyde in PBS (Nacalai Tesque), permeabilized with dilution buffer (0.3% Triton X-100 in PBS containing 3% BSA), and the nuclei were stained with Hoechst 33342. Fluorescent images were captured with the CQ1 Confocal Quantitative Image Cytometer (Yokogawa Electric Corporation, Tokyo, Japan) and the GFP-positive cell numbers were determined using the high-content analysis software, CellPathfinder (Yokogawa Electric Corporation).

### 2.8. Statistical Analysis

GraphPad Prism 9 (GraphPad, San Diego, CA, USA) was used for all statistical analyses. Statistical significance was analyzed by one-way ANOVA and statistically significant differences were determined by Dunnett’s multiple comparisons test (*, *p* < 0.05, significant; **, *p* < 0.01, very significant; ***, *p* < 0.001, highly significant; ns, *p* > 0.05, not significant). Data represent the mean ± standard deviation (S.D.) of at least three independent experiments.

## 3. Results and Discussion

The remarkably low proportions of COVID-19 cases and deaths in East Asian countries compared with those in the USA and Europe have raised the possibility that SNPs specific to East Asian populations negatively affect ACE2-dependent infection with SARS-CoV-2. This is hypothesized to contribute, at least in part, toward the low COVID-19 morbidity and mortality in this region. To investigate this possibility, we searched the Genome Aggregation Database (gnomAD), the Japanese Multi Omics Reference Panel (jMorp) [11], and the 1000 Genomes Project [12] databases to identify SNPs in the *ACE2* gene, causing amino acid changes at the protein level. We selected five SNPs with a relatively high allele frequency of >0.001 (>0.1%) that are found in East Asian or Japanese populations but not in American or European (non-Finnish) populations (Table 1). For comparison, three SNPs not found in East Asian populations were incorporated in our analysis (Table 1). These included two SNPs, S19P and K26R, which are reported to potentially affect ACE2 affinity toward SARS-2-S [9,10], and another SNP, N720D, which is located far from the SARS-2-S binding domain and is, therefore, expected to have little impact on SARS-2-S binding [10]. The amino acid positions corresponding to the selected SNPs were distantly mapped from the region previously reported to be important for binding to the SARS-CoV S protein [16] but are considered to also be important for binding to SARS-2-S based on data from the crystal structure of the ACE2 and SARS-2-S complex [17] (Figure 1). An ACE2 variant with the L656X nonsense mutation in the C-terminus upstream of the transmembrane domain is assumed to not be retained on the cell surface and would, therefore, be incapable of engaging as a cell entry receptor for SARS-CoV-2.

To facilitate investigations of the impacts of the selected SNPs on cell entry mediated by SARS-2-S in vitro, we constructed plasmids expressing C-terminally FLAG-tagged human ACE2 proteins containing the selected SNPs (Figure 2A). First, we examined whether the selected SNPs significantly altered ACE2 protein expression. For this, HEK293T cells were transfected with plasmids expressing the ACE2 variants, and the ACE2 expression levels in the total cell lysates were examined by western blotting (Figure 2B). Consistent with previous observations [18], the HEK293T cells transfected with an empty plasmid (no ACE2 insert) did not exhibit a detectable level of endogenous ACE2. All of the selected SNPs, except for L656X, showed similar overall ACE2 expression levels. As expected, the expression level of the truncation mutant, ACE2(L656X), which cannot be retained on the cellular membrane, was markedly reduced. In general, the cell-surface expression of a receptor molecule is critical as only the portion of the receptor protein expressed on the cell surface can be exploited as an entry receptor. Therefore, we next asked whether the selected SNPs affected the cell-surface expression of ACE2. The cell-surface proteins of HEK293T cells transfected with plasmids expressing WT ACE2 or the ACE2 variants, except for ACE2(L656X), were biotinylated and the total cell lysate was prepared. We omitted the ACE2(L656X) variant from this assay because it showed markedly reduced overall expression (Figure 2B). The biotinylated cell-surface proteins were pulled down (PD) with streptavidin-conjugated magnetic beads. The levels of the cell-surface ACE2 variants in these PD samples were examined by western blotting. No significant differences in the cell-surface protein levels were observed between WT ACE2 and any of the ACE2 variants (Figure 2C).

Although the selected SNPs are not in the amino acid residues critical for SARS-S [16] or SARS-2-S [17] binding (Figure 1), molecular dynamics simulations have indicated that some of these SNPs may potentially affect the affinity of ACE2 toward SARS-2-S. We next examined whether the selected SNPs indeed altered the affinity toward SARS-2-S in vitro. For this, HEK293T cells transfected with plasmids expressing WT ACE2 or the ACE2 variants were incubated with recombinant SARS-2-S1 containing the D614G mutation [rSARS-2-S1(D614G)], a mutation that is currently found in the most prevalent strains of SARS-CoV-2 worldwide [19]. The proportions of cells binding to rSARS-2-S1(D614G) were then analyzed by flow cytometry (Figure 2D). In contrast to recent reports [9,10,20,21], the cells expressing the ACE2 variants, except for ACE2(L656X), bound to rSARS-2-S1(D614G) with similar efficiency to those expressing WT ACE2. Almost no rSARS-2-S1(D614G) binding was observed in cells transfected with the ACE2(L656X) expression plasmid, which was probably due to the negligible level of protein expression for this mutant. Next, we asked whether the selected SNPs affected viral cell entry mediated by SARS-2-S containing the D614G substitution. For this, we used a well-established VSV-based pseudovirus, for which cell entry is solely dependent on compensatory exogenous viral glycoproteins. The VSV pseudotype bearing SARS-2-S has been used widely and reproduces the cell entry process of infectious SARS-CoV-2 effectively [1,22,23]. HEK293T cells transfected with plasmids expressing the ACE2 variants were inoculated with rVSV∆G-GFP/SARS-2-S(D614G) and the proportions of virally infected (GPF-positive, GFP^+^) cells were determined with a high-content imaging system (Figure 2E). Several ACE2 variants showed significantly but only slightly lower efficiency of viral cell entry mediated by SARS-2-S(D614G) and these variants were among both the East Asian or Japanese population-specific (V184A, I468V, and N638S) SNPs and American or European population-specific (K26R and N720D) SNPs. The expression of ACE2(L656X) did not confer viral cell entry into HEK293T cells, most likely because of the low expression of ACE2(L656X). Taken together, these data indicate that the selected SNPs, except for the L656X mutant that abrogated ACE2 expression in the cells, did not affect ACE2 binding to SARS-2-S(D614G). Therefore, these SNPs have limited impact on viral cell entry mediated by SARS-2-S(D614G) and are thus unlikely to contribute toward the varying morbidity and mortality of SARS-CoV-2 observed among different regions.

L656 is located within the cleavage site of the disintegrin and metalloproteinase domain-containing protein 17 (ADAM17). Recent reports indicate that ACE2 cleaved by ADAM17 is released into the extracellular space [24] and this soluble form of ACE2 interacts with SARS-2-S on cell-free SARS-CoV-2 virions, leading to the inhibition of viral infection [25]. These findings and the very low intracellular levels of ACE2(L656X) observed in the present study (Figure 2B) led us to explore the possibility that the L656X mutation promoted rapid ACE2 secretion, resulting in barely detectable intracellular levels of ACE2(L656X) and high levels of shedded ACE2(L656X) that could strongly inhibit SARS-CoV-2 infection. To investigate this possibility, we examined the capacity of the ACE2 variants for shedding into the extracellular space and the subsequent neutralization of viral entry mediated by SARS-2-S. For this purpose, rVSV∆G-GFP/SARS-2-S(D614G) was pre-incubated with serially diluted TCS from HEK293T cells transfected with plasmids expressing WT ACE2 or the ACE2 variants and then inoculated into HEK293T/ACE2 cell cultures. Virally infected (GFP^+^) cell numbers were determined with a high-content imaging system (Figure 3). Unexpectedly, the TCS from cells transfected with the ACE2(L656X) expression plasmid showed no inhibitory effect on SARS-2-S-mediated cell entry, whereas the other ACE2 variants shedded into the TCS did inhibit rVSV∆G-GFP/SARS-2-S(D614G) infection with an efficiency similar to that of WT ACE2 (Figure 3). These results indicate that L656 is essential for ACE2 expression and ACE2 may need to be cleaved by ADAM17 on the cellular membrane to shed soluble ACE2 and inhibit SARS-CoV-2 infection. Intriguingly, the *ACE2* gene is encoded on the X chromosome, and therefore, our results for ACE2(L656X) imply that a male with an L656X SNP would be naturally resistant to SARS-CoV-2.

In the present study, we analyzed the contribution of population-specific SNPs of *ACE2* toward SARS-CoV-2 infectivity in vitro. We showed that none of these SNPs, except for L656X, significantly altered ACE2 binding to SARS-2-S, while only some of these SNPs slightly reduced the efficiency of viral entry mediated by SARS-2-S. As the magnitude of the reduction in viral entry efficiency was small for these SNPs, and because these SNPs are found in both Japanese or East Asian populations and European or American populations, our data indicate that the SNPs analyzed in this study, except for L656X, do not contribute toward the observed differences in the infectivity and severity of COVID-19 between these regions. In addition to ACE2, other host factors, such as TMPRSS2 for S protein priming and the most recently identified cell entry receptor for SARS-CoV-2, neuropilin-1, play critical roles in SARS-CoV-2 multiplication [1,26,27]. Recent in silico studies on TMPRSS2 indicated that several TMPRSS2 SNPs potentially affect SARS-CoV-2 infectivity and contribute toward COVID-19 severity [6,7]. These accumulating in silico findings and the newly identified host factors found to be critical for SARS-CoV-2 multiplication emphasize the importance of future studies to investigate the contribution of SNPs in other host factors toward SARS-CoV-2 infectivity in vitro.

## Figures and Tables

**Figure 1 viruses-13-00067-f001:**
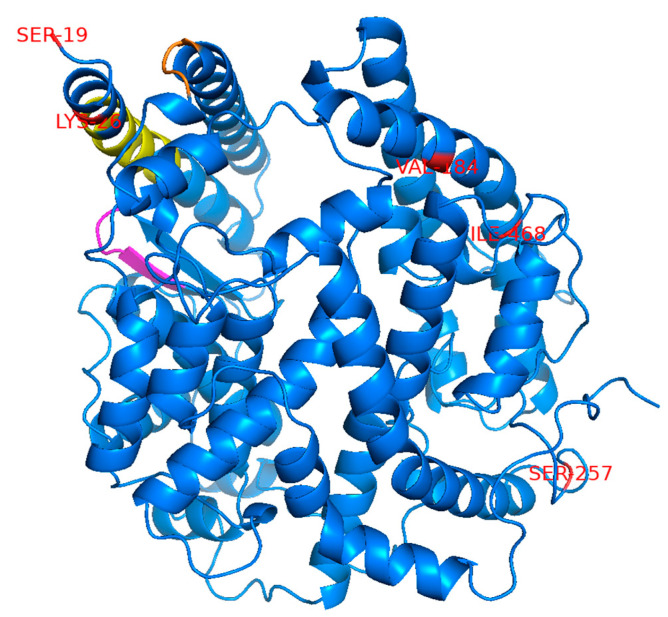
Mapping of the amino acid residues corresponding to the selected SNPs used in this study. The positions of the amino acid residues corresponding to the selected SNPs are marked in red on the crystal structure of ACE2 (1R42). Three amino acid positions (N637, L656, and N720) are not shown here because the regions containing these amino acids are not present in this crystal structure. The three separate regions reported to be important for binding to SARS-CoV-S are colored in yellow, magenta, and orange.

**Figure 2 viruses-13-00067-f002:**
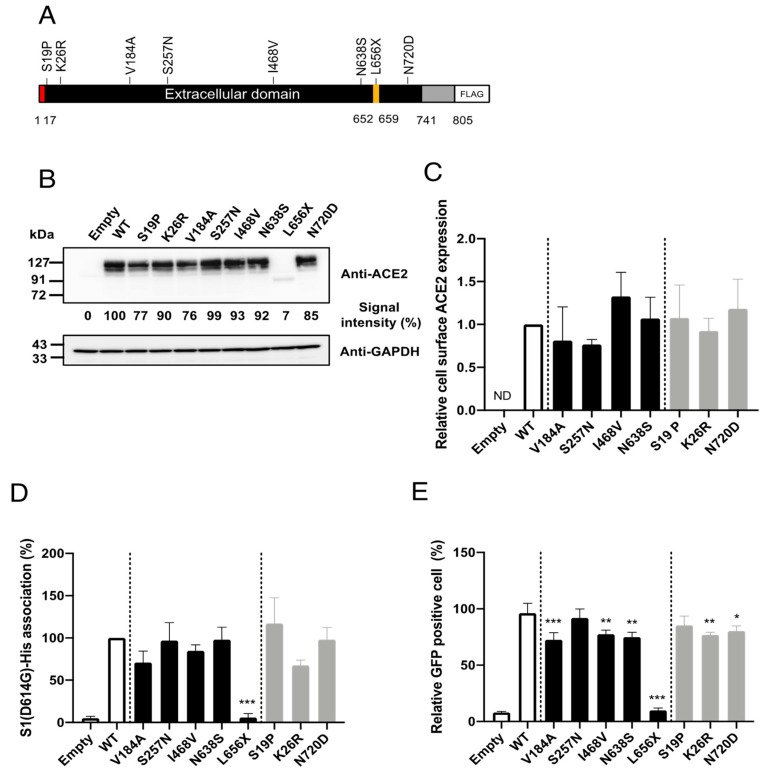
Effects of amino acid substitutions corresponding to SNPs in *ACE2* on binding to SARS-2-S and cell entry mediated by SARS-2-S in vitro. (**A**) Schematic diagram of the construct used to express the ACE2 variants in this study. Amino acid substitutions corresponding to the SNPs were independently introduced into C-terminally FLAG-tagged human ACE2. Red, signal peptide; yellow, ADAM17 cleavage site; gray, transmembrane and cytoplasmic regions. (**B**) HEK293T cells were transfected with plasmids encoding either FLAG-tagged WT or variant ACE2 proteins. The pCMV6-Entry vector only (Empty) was used as the negative control. ACE2 and GAPDH levels in the total cell lysates were analyzed by western blotting. (**C**) HEK293T cells transfected with plasmids expressing either FLAG-tagged WT or variant ACE2 proteins were biotin-labeled and the cell-surface proteins were pulled down with streptavidin beads. The levels of WT ACE2 and ACE2 variants on the cell surface (in the pulled down samples) and in the total cell lysate were then analyzed by western blotting. The levels of transferrin receptor 1 (TFRC) on the cell surface and in the total cell lysate were also analyzed as an internal control. The ratios of the levels of the protein on the cell surface to that in the total cell lysate were calculated for the WT and each variant after first normalizing the data to the corresponding ratios for TFRC. The value for WT ACE2 was set to 1. (**D**) HEK293T cells transfected with plasmids expressing either FLAG-tagged WT or variant ACE2 proteins were incubated with rSARS-2-S1(D614G), and the levels of rSARS-2-S1(D614G) binding to these cells were analyzed by flow cytometry. The value for WT ACE2 was set to 100%. (**E**) HEK293T cells transfected with plasmids expressing either FLAG-tagged WT or variant ACE2 proteins were inoculated with rVSV∆G-GFP/SARS-2-S(D614G). After 16 h, the proportions of GFP-positive cells were determined with a high-content imaging system. (C–E) Data represent the means + S.D. of at least three independent experiments. White bars, controls (vector-only transfected or WT ACE2); black bars, East Asian population-specific SNPs; gray, non-East Asian population-specific SNPs. Empty, transfected with pCMV6-Entry vector only as the negative control; ND, not detected; WT, wild-type ACE2. *, *p* < 0.05; **, *p* < 0.01; ***, *p* < 0.001.

**Figure 3 viruses-13-00067-f003:**
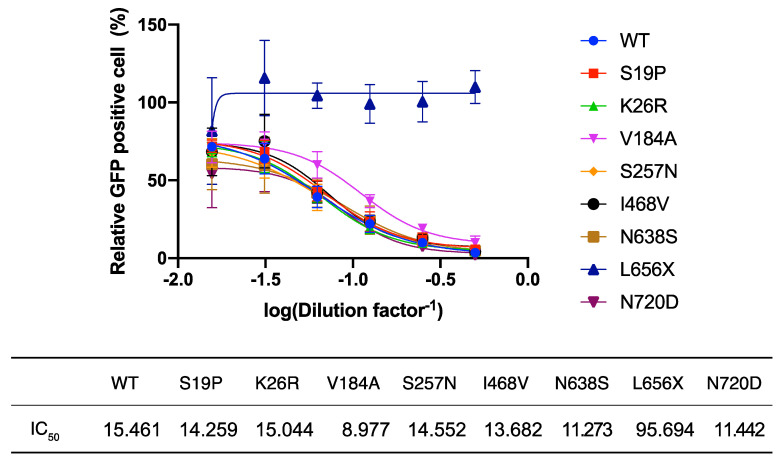
Neutralization assay of shedded ACE2 variants. HEK293T/ACE2 cells were inoculated with rVSV∆G-GFP/SARS-2-S(D614G) pre-incubated for 1 h at 37 °C with serially diluted tissue culture supernatants (TCS) from cells transfected with wild-type (WT) ACE2- or ACE2 variant-expressing plasmids. After 16 h, the levels of virally infected (GFP^+^) cells were measured with a high-content imaging system. The mean number of GFP^+^ cells inoculated with rVSV∆G-GFP/SARS-2-S(D614G) pre-incubated with the TCS from empty vector-transfected cells was set to 100%. The dilution factors required to attain 50% inhibition of viral infection (IC_50_) were determined using GraphPad Prism 9 software. Data represent the mean ± S.D. of three independent experiments.

**Table 1 viruses-13-00067-t001:** ACE2 single-nucleotide polymorphisms (SNPs) found in different populations.

SNPs	Allele Frequencies				
	East Asian *	Japanese (jMorp)	American *	European *	African *
V184A	–	0.0015	0	0	–
S257N	– 0.0013	–	0	0	0
I468V	0.0139 0.0026	0.0035	0	0	0
N638S	0.004 0.0026	–	0	0	0
L656X	– 0.0013	–	0	0	0
S19P	0	–	0	0	0.0031 0.003
K26R	0	–	0.0016 0	0.0056 0.0052	0.0009 0.002
N720D	0	–	0.0032 0.0038	0.0208 0.0183	0.0019 0

* Upper row, data from gnomAD; lower row, data from the 1000 Genomes database.

## Data Availability

The data presented in this study are available on request from the corresponding author.

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
