# Peer review of "Population-Specific ACE2 Single-Nucleotide Polymorphisms Have Limited Impact on SARS-CoV-2 Infectivity In Vitro"

_viruses, 2021, doi:10.3390/v13010067_

Round 1

Reviewer 1 Report

Review of “Population-specific ACE2 single-nucleotide 3 polymorphisms have limited impact on SARS-CoV-2 4 infectivity in vitro” Hashizume et al Viruses

The group of Iwasaki present some much-needed empirical in vitro validation studies to determine the significance of the contribution of population-specific SNPs of ACE2 to SARS-CoV-2 infectivity.  This short communication nicely sums up the current knowledge regarding ACE2 involvement in SARS-CoV-2 infection and provides some initial important data regarding ACE2 SNPs and virus entry (although in a highly artificial system).  

Figure 1 B, the ACE2 runs as a triplet (likely due to glycosylation state).  Does the amount of each form of the ACE2 correlate with any of the differences observed in binding data?  Was any quantitation such as densitometry performed to detect subtle differences in expression efficiency?

What are the proportions of the populations that contain multiple SNP variations?  Any possibility that 2 or more of these mutations in the same person create a synergistic protective effect?

This is obviously a very artificial system with plasmid driven overexpression of ACE2. Are there any comparisons available to endogenous expression of ACE2 on primary lung cells?  Overexpression of proteins can lead to artificially high binding kinetics that wouldn’t be seen under physiological conditions.  Could the small reductions in S/ACE2 binding observed in Figure 1E actually be protective under physiological conditions?  Perhaps including a titration of the plasmid DNA downwards from .2ug with one of the more successful blocking SNPs would show a dose response?

The paper lacks data using wild type SARS-CoV-2 to validate pseudovirus assay results.

 Line 204 “SARS-CoV-2 infectivity” should probably be rewritten as SARS-CoV-2 spike mediated virus entry.

Reviewer 2 Report

I would only suggest to reduce the discussion (is too long); on the other side, the authors may provide some suggestions about the role of this findings on the in vivo and clinical condition useful, with the need of other clinical investigations.

For example, what is the role of some drug medications in the covid-19 patients? 
